# Optimal Infinite Temporal Planning: Cyclic Plans for Priced Timed Automata

**Primary Keywords:** *Temporal Planning*

## Abstract

Many applications require infinite plans—i.e. an infinite sequence of actions—in order to carry out some given process indefinitely. In addition, it is desirable to guarantee optimality. In this paper, we address this problem in the setting of doubly-priced timed automata, where we show how to efficiently compute ratio-optimal cycles for optimal infinite plans. For efficient computation, we present symbolic $\lambda$-deduction ($S$-$\lambda D$), an any-time algorithm that uses a symbolic representation (priced zones) to search the state-space with a compact representation of the time constraints. Our approach guarantees termination while arriving at an optimal solution. Our experimental evaluation shows that $S$-$\lambda D$ outperforms the alternative of searching in the concrete state space, is very robust with respect to fine-grained temporal constraints, and has a very good anytime behaviour.

## 1 Introduction

Temporal planning deals with the problem of finding a (concurrent) plan to achieve a certain goal (Fox and Long 2003; Gigante et al. 2022). However, this disregards the resulting state of the system after executing the plan, and how easy it will be to achieve future goals from there. Many planning applications require instead a cyclic schedule, where a system must stay operational while achieving a goal repeatedly (Draper et al. 1999). Consider for example an assembly line scenario (Asai and Fukunaga 2014) where a large number of units must be produced. A simple approach is to find a plan for producing a single unit and then repeating the plan over and over. However, this may lead to suboptimal plans where no part of the production is parallelised, or even infeasible plans, where after producing some amount the assembly line is left in a state where it cannot produce further units. In such scenarios, we desire infinite plans containing a loop that can be executed arbitrarily many times. Infinite plans have been considered in the context of classical planning with LTL goals (Patrizi et al. 2011), but here we are interested in finding optimal temporal plans with concurrent and delay-constrained actions.

We propose to tackle these applications by finding ratio-optimal cycles in cost-reward timed automata (Bouyer, Brinksma, and Larsen 2004). Thus, assume that the planning problem under consideration may be modelled as a *priced-timed automata* (Behrmann et al. 2001; Alur, Torre,

and Pappas 2004), which has been used to model various real-world scenarios (Hune, Larsen, and Pettersson 2001; Fehnker 1999; Niebert and Yovine 2001; Brinksma, Mader, and Fehnker 2002). The optimal infinite plan has been shown to be a cyclic plan, minimizing the ratio between total cost and reward. This is a very general setting that can, for example, be used to find the schedule that minimizes the monetary cost per produced unit in a factory or that minimizes the amount of time per locations that must be surveyed in a surveillance scenario.

Bouyer, Brinksma, and Larsen (2004) showed that finding a plan with optimal cost-reward ratio is possible using the so-called corner-point abstraction, which reduces the problem to a finite doubly weighted graph by discretizing time. However, this approach requires the entire state space to be enumerated in advance and therefore it scales very poorly with respect to the time scale considered. Thus, practical approaches have focused on finding approximate solutions (e.g., population-based methods (Tolonen, French, and Reynolds 2018)), or analyzing restricted settings, e.g., timed automata with only 1 clock (David et al. 2011).

Our aim is to alleviate the state-space explosion by using *priced zones* (Larsen et al. 2001), a compact representation of sets of states originally introduced in model checking tools for timed automata (Larsen, Pettersson, and Yi 1997). However, this is not trivial in a cost-reward setting, as symbolic representations make it difficult to determine whether there is a concrete cycle as well as determining the cost and reward the concrete paths they abstract over.

We introduce symbolic $\lambda$-deduction ($S$-$\lambda D$), an anytime algorithm that incrementally finds new cycles with improved ratio. To do so, $S$-$\lambda D$ reduces the problem to a single priced automaton and explores the state space symbolically using priced zones, avoiding the enumeration of all possible combinations of clock valuations. We show that the approach converges and terminates with a ratio-optimal cyclic plan, proving that no better solution exists.

We evaluate $S$-$\lambda D$ in three scenarios, showing that it is feasible to compute ratio-optimal cycles in different types of practical applications. $S$-$\lambda D$ has good anytime behaviour, oftentimes finds the optimal cycle faster than the concrete approach, and is far more robust with respect to different time constraints, being almost unaffected by large clock values.

## 2 Background

We adopt the formalism used by Bouyer, Brinksma, and Larsen (2004). A clock is an object that measures the time since it was last reset. A clock *valuation* $u \in \mathbb{R}_{\geq 0}^{\mathbb{C}}$ over the set of clocks $\mathbb{C}$ is a function $u : \mathbb{C} \to \mathbb{R}_{\geq 0}$ assigning a value to each clock. When time passes, it is called a delay, and the value of all clocks increase uniformly. For a delay $\delta \in \mathbb{R}_{\geq 0}$ we denote the updated valuation by $u + \delta$, where $(u + \delta)(x) = u(x) + \delta$. Let $R \subseteq \mathbb{C}$ be a set of clocks to be reset, then $u' = u[R \mapsto 0]$ is the new valuation such that $u'(x) = 0$ if $x \in R$, otherwise, $u'(x) = u(x)$. We denote by $\mathcal{B}(\mathbb{C})$ the set of clock constraints over $\mathbb{C}$ obtained by conjunction over atomic constraints of the type $x \bowtie n$ for $x \in \mathbb{C}, \bowtie \in \{\leq, =, \geq\}$, and $n \in \mathbb{N}$. Let $g \in \mathcal{B}(\mathbb{C})$ be such a constraint, then we write $u \models g$ when $u$ satisfies $g$. For example, for the valuation $u(x) = 3, u(y) = 1$, we have that $u \models x \leq 3$ and $u \not\models y \geq 2 \land x \geq 1$.

**Definition 1.** A *timed automaton* over a set of clocks $\mathbb{C}$ is a tuple $(L, \ell_0, E, I)$, where $L$ is the set of locations, $\ell_0 \in L$ is the initial location, $E \subseteq L \times \mathcal{B}(\mathbb{C}) \times 2^{\mathbb{C}} \times L$ is the set of edges between locations, and $I : L \to \mathcal{B}(\mathbb{C})$ assigns invariants to locations.

A state of the timed automaton consists of a location and a clock valuation. An edge $e = (\ell, g, R, \ell') \in E$ means that the automaton can move from location $\ell$ to $\ell'$ if the clocks satisfy the guard $g$, and after taking this edge the clocks in $R$ will be reset. The invariant of a location $I(\ell)$ is the clock constraints that must be satisfied to be in location $\ell$, thus you cannot move into location $\ell$ if the clock valuation does not satisfy $I(\ell)$ and if already in $\ell$ you cannot delay such that the clock valuation no longer satisfies $I(\ell)$.

The semantics of a timed automaton are given by an underlying labeled transition system $\mathcal{T}_{\mathcal{A}}^* = (\mathcal{S}, s_0, A, T)$, where $\mathcal{S} = L \times \mathbb{R}_{\geq 0}^{\mathbb{C}}$ is the set of concrete states, $s_0 = (\ell_0, \mathbf{0})$ is the initial state, where $\mathbf{0}$ is the valuation mapping all clocks to 0, $A = E \cup \mathbb{R}_{\geq 0}$ is the set of actions, being either an edge or a time to delay, and $T \subseteq \mathcal{S} \times A \times \mathcal{S}$ is the transition relation. We write $s \xrightarrow{\alpha} s'$ whenever $(s, \alpha, s') \in T$ and refer to states in this transition system as concrete. The transition relation then contains delay transitions with $\delta \in \mathbb{R}_{\geq 0}$

$$(\ell, u) \xrightarrow{\delta} (\ell, u + \delta) \text{ if } \forall 0 \leq \delta' \leq \delta.\ u + \delta' \models I(\ell),$$

and edge transitions with $e = (\ell, g, R, \ell') \in E$

$$(\ell, u) \xrightarrow{e} (\ell', u[R \mapsto 0]) \text{ if } u \models g \text{ and } u[R \mapsto 0] \models I(\ell').$$

We now extend timed automata by also including costs and rewards for edges and delays.

**Definition 2.** A *Cost-Reward Timed Automaton* (CRTA) $\mathcal{A} = (L, \ell_0, E, I, c, r)$ is a timed automaton extended with two functions $c, r : L \cup E \to \mathbb{N}_0$, which represent cost and reward rates for locations and edges.

A CRTA $\mathcal{A} = (L, \ell_0, E, I, c, r)$ induces a *Cost-Reward Weighted Transition System* (CR-WTS) $\mathcal{T}_{\mathcal{A}} = (S, s_0, A, T, \text{cost}, \text{reward})$, a labeled transition system extended with functions cost, reward $: T \to \mathbb{R}$, which assign

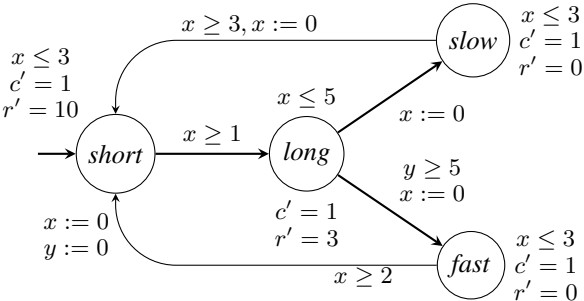

Figure 1: CRTA of a lawnmower example. Resets are denoted by "$x := 0$", cost increments by "$c += n$" (or reward by "$r += n$"), the cost/reward rates of delaying in locations are denoted by "$c' = n$" (and "$r' = n$" for reward). The initial state is marked by a sourceless incoming edge.

cost and reward to transitions.

$$\text{cost}((\ell, u) \xrightarrow{\alpha} (\ell', u')) = \begin{cases} c(\ell) \cdot \delta & \text{if } \alpha = \delta \in \mathbb{R}_{\geq 0} \\ c(e) & \text{if } \alpha = e \in E \end{cases}$$

and dually for reward.

Let $\pi = s_0 \xrightarrow{\alpha_0} s_1 \xrightarrow{\alpha_1} \ldots \xrightarrow{\alpha_{n-1}} s_n$ be a *finite execution* in a CR-WTS $(S, s_0, T, \text{cost}, \text{reward})$ consisting of concrete states. The cost and reward functions extend to finite executions straightforwardly as $\text{Cost}(\pi) = \sum_{i=1}^{n} \text{cost}(s_{i-1}, \alpha_{i-1}, s_i)$ and dually for Reward. We define the cost-reward *ratio* for a finite execution, where $\text{Reward}(\pi) \neq 0$, as $\text{Ratio}(\pi) = \frac{\text{Cost}(\pi)}{\text{Reward}(\pi)}$.

We now consider the case of an *infinite execution*, $\Pi$. Let $\Pi_n$ denote the finite prefix execution of $\Pi$ with length $n$. The ratio of the infinite execution $\Pi$ is then defined by $\text{Ratio}(\Pi) = \liminf_{n \to +\infty} \text{Ratio}(\Pi_n)$, provided this limit exists. The *optimal* ratio for a CR-WTS $\mathcal{A}$ is indicated by $\theta* = \inf \{\text{Ratio}(\Pi) \mid \Pi \text{ is an infinite execution in } \mathcal{A}\}$. An infinite execution $\Pi$ is *ratio-optimal* if $\text{Ratio}(\Pi) = \theta^*$.

We make the restriction that automata must not contain any so-called Zeno-cycles, i.e. all infinite executions must also use an infinite amount of time. Similarly, they must also be *reward-diverging*, meaning all infinite executions that are time-divergent accumulate infinite reward. Finally, we will assume that the automata are bounded, i.e. there exists a $C_{max} \in \mathbb{N}$ s.t. no clock ever has a value larger than $C_{max}$. Behrmann et al. (2001) shows that any unbounded timed automaton can be converted into an equivalent bounded timed automaton, thus no expressiveness is lost.

An interesting subclass of CRTA is when reward corresponds to the time elapsed (i.e., there are no transition rewards and the reward rate at each location is the same). Ratio-optimal plans correspond then to operating the system with lowest cost per time unit. In this case, not having Zeno-cycles is the same as reward-divergence.

**Example 1** Consider the example CRTA shown in Figure 1, which models a lawnmower, whose job is to tend a lawn by keeping it nicely short. When it decides that the grass is too long (after some 1 to 3 time units) it transitions to the *long* location. Here it waits for an additional 2 time units,

but receives less reward. Next, it has the option of whether to mow fast or slow by going to their respective location. The slow approach takes 3 units of time, and the fast approach takes between 2 and 3 units of time. The reward models the quality of the lawn, and the cost models the time. One example of a cycle is to delay 3 units of time in *short*, go to *long*, delay 2, go to *fast*, delay 3, and complete the cycle by returning to *short*. This cycle accumulates 8 cost and 36 reward, and thus it has a ratio of $\frac{8}{36} \approx 0.22$. The optimal plan has a ratio of $\frac{11}{60} \approx 0.18$ and it involves a longer loop that alternates between slow and fast. □

For simplicity we describe our techniques as operating on the full timed automata, as they are agnostic of the exact formalism used. In practice more compact representations can be used. Our implementation is based in UPPAAL(Larsen, Pettersson, and Yi 1997), which uses a representation based on networks of synchronized timed automata. This can be exponentially more compact than the full CRTA, which is only generated in an on-the-fly manner.

### Corner-Point abstraction

Bouyer, Brinksma, and Larsen (2004) introduced *corner-point abstraction*, a method for discretising the clock valuation space by only considering points that are integer-valued. In a bounded timed automata this reduces to a finite state space which can be represented as a doubly weighted finite graph. They showed that this abstraction is sound and complete. There is a slight distinction in our model as, for simplicity, we only allow non-strict clock guards, which simplifies the corner-point abstraction.

The intuition for why it suffices to consider integer valuations only, is that all clock guards have integer bounds and the minimum of a linear fractional function with linear constraints is attained in the corner-points of the zone that the constraints define, hence at integer valuations.

We call a cycle with integer-valued clock valuations a *discrete* cycle. Bouyer, Brinksma, and Larsen (2004) show that there are only finitely many discrete simple cycles and there is always such a cycle that is ratio-optimal.

Wielding the corner-point abstraction, a simple approach is to construct the entire corner-point abstracted state-space as a doubly weighted graph, then use an existing algorithm for the ratio-optimal cycle problem, such as Howard's policy iteration algorithm (Howard 1960; Dasdan, Irani, and Gupta 1999). We use this as a base-line for our symbolic approach.

## 3  Anytime $\lambda$-deduction Algorithm

The main issue with using the corner-point abstraction is that often the size of the graph grows exponentially with the number of clocks. Therefore, constructing the entire graph before starting the search is a huge bottleneck. Instead, we aim to perform an on-the-fly exploration of the state space in order to find the optimal-ratio cycle. However, when dealing with both cost and reward, a problem arises because the costs and rewards of reaching a state can be incomparable. For example, a cost/reward of $1/1$ is not necessarily better or worse than $2/3$ because, depending on the rest of the path to close the cycle, both could be the minimum ratio. If the

rest of the cycle has cost/reward $1/0$, then the better ratio is $3/3$, and if it has $0/2$, then the better ratio is $1/3$. While this could be dealt with multi-objective search (Fränzle et al. 2022; Larsen and Rasmussen 2008), maintaining the entire Pareto front for each clock valuation is unnecessary because we are interested on a single objective: the minimum ratio.

Gondran and Minoux (1995) show how to overcome this problem by combining the cost and reward into a single weight, and then incrementally finding better solutions (according to the current weight) until the optimal is found. This style of algorithm works by maintaining the ratio of the best solution found so far, $\lambda$. The weight at each iteration is $w_\lambda = \mathsf{Cost} - \lambda \cdot \mathsf{Reward}$.

**Proposition 3.** *A cycle $x$ has negative $\lambda$-deducted weight if and only if $\mathsf{Ratio}(x) < \lambda$.*

*Proof.* $\mathsf{Cost}(x) - \lambda \cdot \mathsf{Reward}(x) < 0 \iff \frac{\mathsf{Cost}(x)}{\mathsf{Reward}(x)} < \lambda$. □

This suggests the method used in Algorithm 1: First, pick any cycle $x$ from the CRTA, and let $\lambda := \mathsf{Ratio}(x) = \frac{\mathsf{Cost}(x)}{\mathsf{Reward}(x)}$. Then, find a cycle $x'$ s.t. $w_\lambda(x') < 0$, and update $\lambda := \mathsf{Ratio}(x')$. This is repeated until no such $x'$ exists, at which point an optimal solution has been found.

To use this approach in our setting, we need a method for finding cycles with negative $\lambda$-deducted weight. We therefore introduce a transformation of the original cost-reward automaton into a $\lambda$-deducted *single* priced timed automaton.

**Definition 4.** Let $\mathcal{A} = (L, \ell_0, E, I, \mathsf{c}, \mathsf{r})$ be a CRTA. The $\lambda$-deducted single priced timed automata of $\mathcal{A}$ is $\mathcal{A}_\lambda = (L, \ell_0, E, I, w_\lambda)$, where $w_\lambda(a) = \mathsf{c}(a) - \lambda \cdot \mathsf{r}(a)$ for $a \in L \cup E$.

Note that the weight of any path in $\mathcal{A}_\lambda$ corresponds to the $\lambda$-deducted weight of the path in $\mathcal{A}$. This follows from the semantics of $\mathcal{A}$ and the additive nature of the $\lambda$-deducted weight. We say a cycle in $\mathcal{A}_\lambda$ is a *negative-weight concrete cycle* if the summed weight of the cycle is negative.

Algorithm 1 shows an abstract algorithm for finding ratio-optimal cycles using the $\lambda$-deducted automaton.

**Theorem 5.** *Algorithm 1 terminates, and returns a ratio-optimal concrete cycle, if one exists, otherwise NO CYCLE.*

*Proof.* There are finitely many simple discrete cycles. At each step $\lambda$ decreases. Thus at some point there will be no simple discrete cycle with ratio lower than $\lambda$. □

It is worth noting that, at each iteration, it suffices to find any negative cycle, not necessarily one with minimum (negative) cost. In fact, finding cycles with lower weight does not guarantee faster convergence. For example, with $\lambda = 3$, a cycle with cost/reward $3/2$ has weight $w_\lambda = 3 - 3 \cdot 2 = -3$, while a cycle with a better cost/reward ratio $1/1$ has weight $w_\lambda = 1 - 1 \cdot 2 = -1$.

## 4  Symbolic $\lambda$-Deduction

All that remains to instantiate the $\lambda$-deduction algorithm is an effective and efficient way for finding negative-weight concrete cycles in $\mathcal{A}_\lambda$. This could be done with an explicit search of the corner-point abstracted state space. However,

**Algorithm 1:** $\lambda$-deduction algorithm.

> **input** : A bounded and strongly reward-divergent CRTA
> $\mathcal{A} = (L, \ell_0, E, I, \mathsf{c}, \mathsf{r})$ over the clocks $\mathbb{C}$.
> **output:** A ratio-optimal concrete cycle, if one exists,
> otherwise NO CYCLE.
> 
> **1** **if** $\mathcal{A}$ has no cycle **then**
> **2**     **return** NO CYCLE
> **3** $C_\lambda :=$ any cycle in $\mathcal{A}$
> **4** $\lambda := \mathsf{Ratio}(C_\lambda)$
> **5** **while** $\mathcal{A}_\lambda$ has negative-weight simple discrete cycle $C$ **do**
> **6**     $\lambda := \mathsf{Ratio}(C)$
> **7**     $C_\lambda := C$
> **8** **return** $C_\lambda$

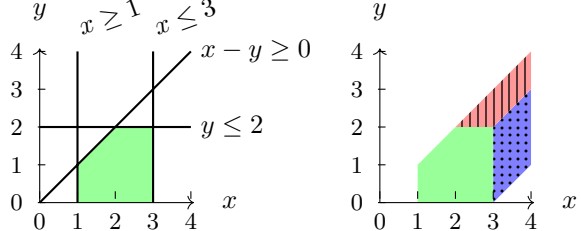

Figure 2: Zone (in green, left) induced by 4 constraints over clocks $x$ and $y$ and successors (right) after a delay transition.

as the number of states grows exponentially with the number of clocks, we utilise a symbolic representation of the clock-space.

## Symbolic Zone-Based Representation

In model checking, *zones* are commonly used as a symbolic representation of a subset of the clock space (Daws et al. 1995; Larsen, Pettersson, and Yi 1997). A zone over a set of clocks $\mathbb{C}$ is defined by a conjunction of constraints, which can be upper or lower bounds on either the value of clocks or the difference between two clocks. Thus, the zone represents a convex subset of $|\mathbb{C}|$-dimensional Euclidean space.

As we are also dealing with a price, namely the $\lambda$-deducted weight, we use *priced zones* (Larsen et al. 2001). Priced zones extend zones with a price function, an affine function from the clock values to the price of reaching that particular state. Figure 2 (left) shows an example of a zone. Assuming that the price function is $w_{green}(u) = -1 \cdot u(x) + 2 \cdot u(y) + 1$, the price of reaching the valuation $(x = 2, y = 1)$ is $w_{green}(2, 1) = -1 \cdot 2 + 2 \cdot 1 + 1 = 1$.

A symbolic state $S = (\ell, Z, w)$ consists of a location, a zone and a price function. We say that a concrete state $s = (\ell, u)$ is in $S$, denoted by $s \in S$, iff $u \in Z$.

Zones themselves are represented using *difference bound matrices*(Dill 1989; Bengtsson and Yi 2003), a data structure for efficient storing and performing necessary operations on zones. We define the functions $\mathsf{Post}_e$ and $\mathsf{Post}_\epsilon$ that return, for a given symbolic state $S = (\ell, Z, w)$, the symbolic successors after using the discrete edge and delay actions, respectively. The delay successors of a symbolic state,

$\mathsf{Post}_\epsilon(S)$ reflect the lowest price of reaching a specific point the successors from some point in $S$. Each delay action has an associated delay rate $r$. The slope of the delay trajectory of a priced zone is the price of increasing all clocks by 1 according to $w$, let us call this $q$. There are only two ways to reach a valuation to consider: (i) delaying as much as possible in this delay action; (ii) delaying the least possible. We should choose (i) when $r \leq q$ and choose (ii) when $r \geq q$.

The result of delaying can be represented as one or more symbolic states. The example of Figure 2 (right) shows the successors after a symbolic delay action on the zone depicted in the left, assuming the location $\ell$ has an invariant $x \leq 4$, and that we delay in a location with weight-rate $r = 3$. In this case, the slope of the delay trajectory of the green zone is $q = 1$ (i.e. the sum of the coefficients for $u(x)$ and $u(y)$); thus we should delay the least possible. Therefore, only the valuations not already reachable from green are represented in separate symbolic states with new zones. The two new zones, red and blue (respectively, vertical lines and dots), have price functions $w_{red}(u) = -1 \cdot u(x) + 1 \cdot u(y) + 2$ and $w_{blue}(u) = 1 \cdot u(x) + 2 \cdot u(y) - 5$. There are similar considerations for the discrete edge successor; we refer the reader to Larsen et al. (2001) for full details.

## Symbolic Cycles

We write $S \xrightarrow{\alpha} S'$ whenever $S' \in \mathsf{Post}_\alpha(S)$ for some $\alpha \in E \cup \{\epsilon\}$. When the particular action does not matter, we simply write $S \rightarrow S'$. A symbolic path $\Pi = S_0 \xrightarrow{\alpha_0} S_1 \rightarrow \cdots \xrightarrow{\alpha_{n-1}} S_n$ is sequence of symbolic transitions. We use the notation $S_0 \rightsquigarrow S_n$ whenever there exists a symbolic path from $S_0$ to $S_n$. We say that a concrete path $\pi$ is in a symbolic path $\Pi$, denoted $\pi \in \Pi$, when $\pi$ and $\Pi$ agree on the discrete transitions and all concrete states in $\pi$ are contained in the corresponding symbolic state in $\Pi$.

**Definition 6.** A symbolic path $\Pi$ is a *symbolic cycle* iff there exists a concrete cycle $\pi \in \Pi$. A symbolic cycle $\Pi$ is a *negative-weight symbolic cycle* iff there exists a negative-weight concrete cycle $\pi \in \Pi$.

We first consider the problem of, given a symbolic path, how to determine whether it is a (negative-weight) symbolic cycle. This is an important question, as we want to symbolically search for negative-weight concrete cycles. However, this is not readily apparent as there may be an overlap in the start and end zones but no concrete cycle from one point back to itself. Furthermore, despite priced zones keeping the price of reaching any point in the end zone, we cannot determine the price of a particular path from start to end. For a given end valuation, we only know the cheapest price of reaching this valuation, but not from which valuation a path with this price is realisable from. Next, we will show how to extract the best concrete cycle from a symbolic path, both w.r.t. to ratio and $\lambda$-deducted weight.

Extracting the best concrete cycle from a symbolic cycle consists of finding optimal concrete delays for the symbolic delays. However, since symbolic states abstract over many concrete states, there are also concrete cycles that do multiple revolutions inside the symbolic cycle. Specifically, given a symbolic cycle $\Pi$, we want to find the best concrete cycle

$\pi$ s.t. there exists a $k$ for which $\pi \in \Pi^k$, where $\Pi^k$ is $\Pi$ concatenated onto itself $k$ times.

**Theorem 7.** *Let $\Pi = (\ell, Z_1) \rightsquigarrow (\ell, Z_n)$ be a symbolic cycle. Then for all $k > 1$, the optimal concrete cycle in $\Pi^1$ has the same ratio as the optimal concrete cycle in $\Pi^k$.*

*Proof sketch.* Let $\pi^k$ be a cycle in $\Pi^k$. We will then show how to construct a cycle $\pi^1 \in \Pi^1$ that has the same ratio. Let $\pi_1 \pi_2 \cdots \pi_k = \pi^k$ be the partitioning of $\pi^k$ s.t. $\pi_i \in \Pi$ for $0 < i \leq k$. We then construct $\pi^1$ as a convex combination of all $\pi_i$. For a scalar $\eta \in \mathbb{R}_{\geq 0}$ and a valuation $u \in \mathbb{R}_{\geq 0}^{\mathbb{C}}$, we define $(\eta \cdot u)(x) = \eta \cdot u(x)$ for $x \in \mathbb{C}$. Also, we define $(u + u')(x) = u(x) + u'(x)$ for $x \in \mathbb{C}$.

Let $\eta = \frac{1}{k}$, we then construct $\pi^1$ as the convex combination, such that each state $(\ell_j, v_j) \in \pi^1$ has $v_j = \sum_{i=1}^{k} u_j^i$ where $(\ell_j, u_j^i) \in \pi_i$. The delays are then also the convex combination of the delays of all $\pi_i$. It can be shown that $\pi^1$ is a valid cycle and has the same ratio as $\pi^k$. Since the zones are convex, $\pi^1 \in \Pi$. The reset operation also works in the convex combination because they all agree that the reset clock is 0. The ratio is the convex combination of all of the ratios, since $\eta = \frac{1}{k}$ it is the same as the ratio of $\pi^k$. $\square$

**Corollary 8.** *Let $\Pi = (\ell, Z_1) \rightsquigarrow (\ell, Z_n)$ be a symbolic cycle in a single weight timed automata. Then for all $k > 1$, there exists a concrete cycle with weight $w$ in $\Pi^1$ if and only if there exists a concrete cycle with weight $w \cdot k$ in $\Pi^k$.*

*Proof.* This follows easily from Theorem 7. We can simply pretend that $\mathsf{Reward}(\Pi) = 1$, then there is a cycle $\pi \in \Pi$ with $\mathsf{Ratio}(\pi) = \frac{w}{1}$ iff there is a cycle $\pi^k \in \Pi^k$ with $\mathsf{Ratio}(\pi^k) = \frac{k \cdot w}{k}$. $\square$

With this, we know that we can limit ourselves to finding cycles of a single revolution in the symbolic cycle; thus, we will now describe how to extract a concrete cycle. We will first focus on extracting the best concrete cycle w.r.t. cost-reward ratio. Tolonen, French, and Reynolds (2018) show how this is achieved by optimisation the delays of the symbolic cycle using linear-fractional programming (Charnes and Cooper 1962); here, we give only an overview of the procedure. We optimise the delays to minimise over the fractional objective that is the cost-reward ratio of the cycle. For a symbolic cycle $\Pi$, the core idea is to construct timestamps $t_i$ for each $\alpha_i$ in $\Pi$, representing the time elapsed since the action $\alpha_0$. Additionally, a timestamp $t_{n+1}$ is added as the end of the cycle, where $n = |\Pi|$, i.e. the duration of the cycle. Recall that zones are constructed from difference constraints on clock values. Therefore, zones can be transformed into constraints on timestamps by translating a clock value at index $i$ into a difference constraint on the current timestamp and the timestamp at which it was last reset. For a constraint $x \leq n$ at index $i$, we find the index $j$ that is the latest index (w.r.t. $i$) where $x$ was reset. Then, if $j \leq i$ the constraint is equivalent to $t_i - t_j \leq n$, and if $j > i$, i.e. if $x$ was only reset in the previous iteration of $\Pi$, then the constraint is equivalent to $t_i + t_{n+1} - t_j \leq n$.

We can therefore determine whether a symbolic cycle is negative by optimising for the lowest weight concrete cycle and seeing if it is negative, or alternatively, optimize for the best cost-reward ratio concrete cycle and checking if it is less than the current $\lambda$ (i.e., the best ratio found so far). The latter is preferred, as we might as well extract the best cost-reward ratio cycle, as this will maximise the improvement of $\lambda$.

## Symbolic Search for Negative Cycles

We are now ready to present our algorithm for finding negative-weight cycles. Algorithm 2 performs best-first search, where each search node corresponds with a symbolic state. We maintain a parent pointer, $Parent$, that maps each discovered symbolic state to its predecessor. Specifically, the $Parent$ pointer returns for $S$ either a pair $(S', \alpha)$, where $S'$ is the predecessor and $e \in E$ is the action, s.t. $S' \xrightarrow{e} S$, or the special symbol NIL when $S$ is the initial state.

On line 5, the algorithm extracts and expands a minimum element from $Waiting$, according to some ordering. The order chosen has no consequence on termination and correctness of the algorithm, but it may affect the efficiency. We suggest to choose a state containing the minimum weight valuationfrom the intuition that cheap valuations are more likely to produce negative cycles.

Every time a new successor is generated, we check whether it belongs to a negative-weight cycle (line 10) by calling the subroutine FIND-NEGATIVE-CYCLE. This recursive subroutine follows the $Parent$ pointer backwards, checking whether the suffix is a cycle. This is done with the linear-fractional program for all symbolic paths that satisfy some necessary conditions (e.g. there is some concrete state $s$ such that $s \in S_1$ and $s \in S_n$). Thus, FIND-NEGATIVE-CYCLE returns a negative-weight symbolic cycle if one is found, otherwise it will eventually reach NIL and return NO CYCLE.

The key problem is when to stop searching. It may seem that duplicate detection is safe, i.e., if we return to a state $(\ell, Z, w)$ with the same location and zone, there is no point in continuing the search. However, that is not the case. Duplicate pruning ignoring $w$, may incorrectly prune the optimal cycle, which may pass multiple times by the same symbolic state.

**Proposition 9.** *There exist CRTAs such that the optimal concrete cycle is contained in a non-simple symbolic cycle.*

*Proof sketch.* Consider the automaton:

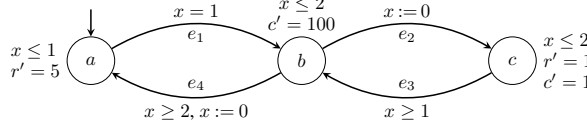

The optimal concrete cycle is $(a, 0) \xrightarrow{1} (a, 1) \xrightarrow{e_1} (b, 1) \xrightarrow{e_2} (c, 0) \xrightarrow{2} (c, 2) \xrightarrow{e_3} (b, 2) \xrightarrow{e_4} (a, 0)$, with a ratio of $\frac{2}{7}$. However, this is in the non-simple symbolic cycle $S_0 \rightarrow S_1 \rightarrow S_2 \rightarrow S_3 \rightarrow S_4 \rightarrow S_5 \rightarrow S_3 \rightarrow S_0$, with both $S_0$ and $S_3$ appearing twice. There are two simple symbolic cycles $S_0 \rightarrow S_1 \rightarrow S_2 \rightarrow S_3 \rightarrow S_0$ and $S_3 \rightarrow S_4 \rightarrow S_5 \rightarrow S_3$, however, the optimal concrete cycles in these only achieve a cost-reward ratio of $\frac{100}{5}$ and $\frac{1}{1}$, respectively. $\square$

Note that this does not contradict Theorem 7, as both parts of the cycle are using different edges.

---

**Algorithm 2:** Find Negative Cycle of $S$-$\lambda D$.

---

**input** : A single priced timed automaton
        $\mathcal{A}_\lambda = (L, \ell_0, E, I, w_\lambda)$.

**output:** A negative weight simple discrete cycle, if one
        exists, otherwise NO CYCLE.

---

1   **Function** FIND-NEGATIVE-CYCLE()
2      $Waiting := \{\, S_0 \,\}$ // initial state
3      $Parent := \{S_0 \mapsto \text{NIL}\}$
4      **while** $Waiting \neq \emptyset$ **do**
5         $S := \text{EXTRACT-MIN}(Waiting)$
6         **forall** $S' \in Post_\epsilon(\text{Post}_e(S))$ for all $e \in E$ **do**
7            **if** $\forall S'' \in \text{dom}(Parent).\ S'' \not\sqsubseteq S'$ **then**
8               $Parent(S') := (S, \alpha)$
9               insert $S'$ into $Waiting$
10               **if** NEG-CYCLE $(S \xrightarrow{\alpha} S')$ returns cycle $C$ **then**
11                  **return** $C$

12      **return** NO CYCLE

13   **Function** NEG-CYCLE $(S_1 \xrightarrow{\alpha_2} \cdots \xrightarrow{\alpha_n} S_n)$
14      **if** $S_1 \xrightarrow{\alpha_2} \cdots \xrightarrow{\alpha_n} S_n$ is a negative-weight cycle **then**
15         **return** best concrete cycle in $S_1 \xrightarrow{\alpha_2} \cdots \xrightarrow{\alpha_n} S_n$
16      **if** $Parent(S_1) \neq$ NIL **then**
17         $(S, \alpha) := Parent(S_1)$
18         **return** NEG-CYCLE $(S \xrightarrow{\alpha} S_1 \xrightarrow{\alpha_2} \cdots \xrightarrow{\alpha_n} S_n)$
19      **else**
20         **return** NO CYCLE

---

To ensure termination, we define a domination criteria to prune symbolic states if they have a higher price everywhere.

**Definition 10.** A priced symbolic state $S = (\ell, Z, w)$ *dominates* another $S' = (\ell', Z', w')$, denoted by $S \sqsubseteq S'$, iff (i) $\ell = \ell'$, (ii) $Z \supseteq Z'$, and (iii) $w(u) \leq w'(u)$ for all $u \in Z'$.

When searching, we do not expand a state $S$ if we have already seen a state $S'$ where $S' \sqsubseteq S$. On line 7, only successors where no dominating state has already been discovered are added to $Waiting$. This ensures that an optimal cycle will not be pruned, and it is sufficient to ensure that the algorithm terminates. If prices keep becoming lower and lower, then a negative cycle will be found. If there is no negative cycle, eventually no non-dominated state will be left, and the algorithm will return NO CYCLE.

**Lemma 11.** *The subroutine* FIND-NEGATIVE-CYCLE *in Algorithm 2 terminates.*

*Proof.* We first argue that there exists a constant $w_{min}$ for $\mathcal{A}_\lambda$ s.t. any symbolic path that contains a concrete path with weight less than $w_{min}$ must contain a negative-weight concrete cycle. Let $\mathcal{T}_{cp} = (\mathcal{S}_{cp}, s_0, A, T_{cp}, \text{cost}, \text{reward})$ be the CR-WTS induced by the corner point abstraction of the automaton $\mathcal{A}$, i.e. containing only discrete states and transitions, then $w_{min} = \sum \{\, w_\lambda(t) \mid t \in T_{cp}, w_\lambda(t) < 0 \,\}$, where $w_\lambda(t) = \text{cost}(t) - \lambda \cdot \text{reward}(t)$, i.e. the sum of the weight of all negative $\lambda$-deducted transitions. Any concrete

path with weight less than $w_{min}$ must use at least one negative edge more than once, and therefore, it contains at least one concrete cycle. At least one of these cycles must be negative.

Now, let $\mathcal{S}$ be the set of all priced symbolic states and $\mathcal{S}_{\geq w_{min}} = \{\, (\ell, Z, w) \in \mathcal{S} \mid \forall u \in Z.\ w(u) \geq w_{min} \,\}$. We now argue that $(\mathcal{S}_{\geq w_{min}}, \sqsubseteq)$ is a well-quasi order, i.e. there exists no infinite descending sequence where the states are not dominated by a previous state. For a bounded priced timed automaton, there is only a finite number of (unpriced) zones, thus a zone must eventually repeat in an infinite sequence. The minimum of a priced zone is attained in an integer point, therefore, we can limit ourselves to study the cost plane only in the finite number of integer points. Let $u \in Z$ be an integer point of zone $Z$, $\pi$ be a minimum $\lambda$-deducted weight path to $u$, and $\lambda = \frac{a}{b}$ for $a \in \mathbb{Z}$ and $b \in \mathbb{N}^+$. Let $\mathbb{Q}_{\frac{1}{b}} = \{\, q \in \mathbb{Q} \mid \exists c \in \mathbb{Z}.\ q = \frac{c}{b} \,\}$ be the set of rationals that can be written with $b$ as denominator. The $\lambda$-deducted weight of reaching the integer point is $w_\lambda(\pi) = \text{Cost}(\pi) - \frac{a}{b} \cdot \text{Reward}(\pi) = \frac{\text{Cost}(\pi) \cdot b - a \cdot \text{Reward}(\pi)}{b}$, and since $\text{Cost}(\pi)$ and $\text{Reward}(\pi)$ are integer-valued (known from corner-point abstraction), $w_\lambda(\pi) \in \mathbb{Q}_{\frac{1}{b}}$. For all $b \in \mathbb{Z}^+$, $\mathbb{Q}_{\frac{1}{b}}$ is nowhere dense in $\mathbb{R}$ and thus any infinite descending sequence must tend towards negative infinity. Thus, the weight of the integer points cannot keep improving ad infinitum, as it will become less than $w_{min}$, and since there are only finitely many integer points eventually no point can improve further. This, in conjunction with the finiteness of zones, shows that $(\mathcal{S}_{\geq w_{min}}, \sqsubseteq)$ is a well-quasi order. The argument is similar to Dickson's Lemma (Kruskal 1972).

These properties guarantee that our algorithm terminates. Firstly, the subroutine FIND-NEGATIVE-CYCLE terminates either when $Waiting = \emptyset$ or when NEG-CYCLE returns a cycle. The if-statement on line 7 ensures that we never added a state to $Waiting$ that is dominated by a previously expanded state. Thus only a finite amount of dominated states will be expanded. If no state is ever expanded that contains a point cheaper than $w_{min}$, then there exists no infinite descending sequence for $\sqsubseteq$, and thus the algorithm must terminate. Otherwise, whenever any state containing a point cheaper than $w_{min}$ is expanded, then there must be a negative cycle present in the parent pointer, so therefore it terminates. The subroutine NEG-CYCLE is called whenever a state is added to $Waiting$, thus when the parent pointer contains a cycle, it is detected. We will assume that the linear programming solver finds a discrete cycle. However, if it does not, this does not affect convergence of the algorithm, as we know from corner-point abstraction (Bouyer, Brinksma, and Larsen 2004) that there exists an equivalent discrete cycle with the same cost-reward ratio. $\square$

Symbolic $\lambda$-deduction, i.e., Algorithm 1 using Algorithm 2 as a sub-routine in line 5 is complete, sound, and optimal.

**Theorem 12.** *The symbolic $\lambda$-deduction algorithm terminates, and it returns a ratio-optimal concrete cycle, if one exists, otherwise* NO CYCLE.

*Proof sketch.* We know that Algorithm 1 is correct and

terminates (Theorem 5), therefore, it suffices to show that FIND-NEGATIVE-CYCLE is correct and terminates. We handled termination already in Lemma 11. Here we prove the correctness as soundness and completeness.

*Soundness:* if FIND-NEGATIVE-CYCLE returns a cycle then it is a negative-weight concrete cycle in $\mathcal{A}_\lambda$. The algorithm only expands reachable states and only returns a cycle if it is negative. The concrete cycles are extracted by linear program optimisation, and by Corollary 8, we need only consider one revolution in each symbolic cycle. Thus, the cycle is reachable and has negative-weight.

*Completeness:* if there exists a negative cycle in $\mathcal{A}_\lambda$ then FIND-NEGATIVE-CYCLE returns a cycle. Recall from the proof of Lemma 11, that if there exists a negative cycle, then there is a weight $w_{min}$ s.t. reaching a weight below this can only be done with a negative cycle. Also, a negative cycle always makes the weight arbitrarily cheap. From the priced zones, we know that there is always a symbolic path to reach any reachable state that is descending w.r.t. $\sqsubseteq$; i.e. no symbolic state is dominated by a preceding state in the path. Thus, the descending symbolic path to reach the state with weight less than $w_{min}$, must eventually be found by the algorithm, and this must contain a concrete cycle. Thus the symbolic cycle will be discovered because we check of a negative cycle every time a state is added to *Waiting*.  $\square$

## 5  Experimental Evaluation

We conduct experiments comparing symbolic $\lambda$-deduction (*S-$\lambda$D*) against the baseline *CP-MCR*. *CP-MCR* constructs the entire corner-point abstracted concrete state space and then uses existing techniques to find a ratio optimal cycle in the doubly weighted graph. We implemented both algorithms on top of UPPAAL 4.1 (Larsen, Pettersson, and Yi 1997). *CP-MCR* uses the Boost library's solver for minimum cycle ratio problems, which is based on Howard's algorithm (Dasdan, Irani, and Gupta 1999). All experiments were run on a cluster with AMD EPYC 7551 32-core processors, with a 10-minute time limit and 10GB memory limit. The data from the experiments and the scripts for generating the figures and tables will be made publicly available.

We conduct experiments in three different domains: surveillance, job scheduling and volunteer. The problems are modelled in UPPAAL as networks of timed automata. Larsen, Pettersson, and Yi (1997) gives a thorough description of how modelling in UPPAAL works. We extended UPPAAL's parsing module to allow for both costs and rewards.

**Surveillance**  In this domain one or more agents must surveil different places. If a place has not been surveilled for 10 units of time, the cost of the place is increased. The agents can either be waiting or surveilling, with the former giving more reward. At some point the agent can choose to surveil a place, this takes between 5 to 10 units of time and resets the cost of the place to its initial value. We denote instances of this problem as $S_{a,p}$, where $a$ is the number of agents and $p$ is the number of places.

**Job Scheduling**  This is a small extension to the scheduling problem by Behrmann, Larsen, and Rasmussen (2005).

| Inst. | CP-MCR | | | | S-$\lambda$D | | | |
|---|---|---|---|---|---|---|---|---|
| | $t_{opt}$ | $t_s$ | $t_f$ | States | $t_{opt}$ | $t_s$ | $t_f$ | States |
| $J_{2,1}$ | **0.00** | **0.00** | **0.00** | 173 | **0.00** | **0.00** | 0.00 | 12 |
| $J_{2,2}$ | 0.29 | 0.23 | 0.23 | 20k | **0.07** | **0.03** | 0.00 | 1k |
| $J_{2,3}$ | 252.29 | 232.00 | 230.86 | 2M | **134.51** | **0.03** | **0.02** | 225k |
| $J_{2,4}$ | OOT | - | - | - | OOT | - | 0.09 | - |
| $J_{3,2}$ | 0.54 | 0.35 | 0.35 | 27k | **0.06** | **0.01** | 0.00 | 1k |
| $J_{3,3}$ | OOT | - | - | - | **494.80** | **0.65** | **0.16** | 357k |
| $J_{3,4}$ | OOT | - | - | - | OOT | - | 4.65 | - |
| $S_{1,1}$ | **0.00** | **0.00** | **0.00** | 78 | **0.00** | **0.00** | 0.00 | 7 |
| $S_{1,2}$ | 0.04 | 0.03 | 0.03 | 3k | **0.00** | **0.00** | 0.00 | 119 |
| $S_{1,3}$ | 1.70 | 1.48 | 1.37 | 63k | **0.21** | **0.01** | **0.00** | 3k |
| $S_{1,4}$ | 70.81 | 64.71 | 63.78 | 636k | **11.35** | **0.01** | **0.01** | 71k |
| $S_{1,5}$ | OOT | - | - | - | **54.22** | **0.03** | **0.03** | 213k |
| $S_{1,6}$ | OOT | - | - | - | **49.94** | **3.51** | **3.51** | 373k |
| $S_{1,7}$ | OOT | - | - | - | **87.44** | N/A | - | 702k |
| $S_{2,1}$ | 0.01 | 0.01 | 0.01 | 1k | **0.00** | **0.00** | 0.00 | 39 |
| $S_{2,2}$ | 2.15 | 1.84 | 1.80 | 50k | **0.14** | **0.00** | **0.00** | 1k |
| $S_{2,3}$ | OOT | - | - | - | **93.83** | **0.01** | **0.01** | 102k |
| $S_{2,4}$ | OOT | - | - | - | OOT | - | 0.01 | - |
| $S_{3,1}$ | 0.35 | 0.29 | 0.28 | 14k | **0.04** | **0.00** | **0.00** | 629 |
| $S_{3,2}$ | 320.94 | 313.35 | 312.51 | 770k | **44.15** | **0.00** | **0.00** | 50k |
| $S_{3,3}$ | OOT | - | - | - | OOT | - | **0.00** | - |
| $V_{1,1,1}$ | **0.03** | **0.02** | 0.02 | 1k | 0.14 | 0.10 | **0.00** | 4k |
| $V_{1,1,2}$ | **1.23** | **1.15** | 0.61 | 20k | 33.91 | 18.74 | **0.00** | 143k |
| $V_{1,1,3}$ | **363.67** | **359.67** | 123.73 | 542k | OOT | - | **0.25** | - |
| $V_{1,2,1}$ | **5.07** | **3.97** | 2.83 | 60k | 62.32 | 27.46 | **0.00** | 235k |
| $V_{1,2,2}$ | **320.92** | 287.45 | 279.12 | 694k | OOT | - | **0.02** | - |
| $V_{1,2,3}$ | OOT | - | - | - | OOT | - | **0.59** | - |
| $V_{2,1,1}$ | **0.23** | **0.20** | 0.16 | 10k | 0.54 | 0.43 | **0.00** | 20k |
| $V_{2,1,2}$ | **12.97** | **12.06** | 5.02 | 115k | 253.93 | 173.78 | **0.00** | 1M |
| $V_{2,1,3}$ | OOT | - | - | - | OOT | - | **0.28** | - |

Table 1: Results of comparison between *S-$\lambda$D* and *CP-MCR* in terms of time in seconds until finding the first (non-optimal) solution ($t_f$), the optimal solution without proving optimality ($t_s$), and until termination ($t_{opt}$). OOT means that the time limit was exceeded (the memory limit was never exceeded). The States column indicates the number of concrete (for *CP-MCR*) and symbolic states (for *S-$\lambda$D*).

We made it cyclic, adding a reward for completing jobs. The model concerns with scheduling a set of jobs on specific machines. Each job needs to serially perform two tasks on two specific machines for an amount of time, and it has a frequency at which it needs to be repeated (e.g., job $A$ needs to use 10s on machine 1 and then 5s on machine 3, and the job must be completed every 60 seconds). Completing the task provides reward and running the machines has a cost. We denote instances of this problem as $J_{m,j}$, where $m$ is the number of machines and $j$ is the number of jobs.

**Volunteer**  This is a scheduling problem which concerns volunteers balancing their university work while maintaining a supply of cold drinks in the refrigerators. The refrigerator can store up to 4 items, which can be withdrawn by the consumers and volunteers, yielding reward. A volunteer can decide to walk to the refrigerators to refill them, this adds to the cost and spends time. The consumers start as active members, but become inactive if they do not withdraw an

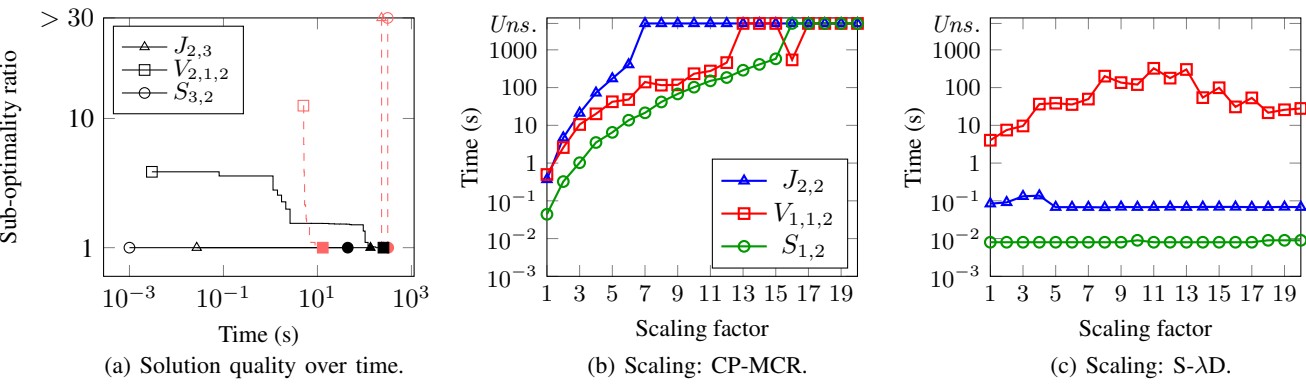

(a) Solution quality over time.

(b) Scaling: CP-MCR.

(c) Scaling: S-λD.

Figure 3: 3a shows the sub-optimality ratio over time for *S-λD* (black, solid) and *CP-MCR* (red, dashed). Figures 3b and 3c show the time of *CP-MCR* and *S-λD* on $J_{2,2}$, $V_{1,1,2}$ and $S_{1,2}$ when scaling all clock constants.

item within a given time interval. An inactive consumer no longer withdraws items and does not generate any reward. The volunteers earn reward by working in their group room, as well as when they withdraw items. We denote instances as $V_{f,v,c}$, where $f$ is the number of refrigerators, $v$ is the number of volunteers, and $c$ is the number of consumers.

We first analyze scalability with respect to the number of objects, i.e., the number of concurrent timed automata and thus the number of locations and clocks in the full CRTA. All clock constants in invariants and guards are small ($\leq 15$), and are kept constant across all instances of the domain. Table 1 shows the results. Overall, we see that *S-λD* clearly outperforms *CP-MCR* in the job and surveil domains, solving more problems and being significantly faster and with lower memory usage. We also note that on $S_{1,7}$, symbolic $\lambda$-deduction finds that there is no cycle, as a single agent cannot surveil seven places within the time constraints. However, *CP-MCR* outperforms *S-λD* on volunteer. This is partially due to a larger number of iterations (up to 22 in $V_{2,1,2}$) finding improving cycles, and due to the large length of the optimal cycle (up to 416/7118 steps in $V_{1,1,2}/V_{1,1,3}$, where as in other domains are only up to 25 steps). Contrary to *CP-MCR*, *S-λD* uses different states in each iteration, leading to nearly an order of magnitude more states, even though *S-λD*'s states are using priced zones and thus correspond to multiple explicit states. The number of states gives a clear indication of why *S-λD* performs poorly on the volunteer domain. *S-λD* searches nearly an order of magnitude more states, even though *S-λD*'s states are using priced zones and thus correspond to multiple explicit states.

An important advantage of *S-λD* is its good anytime behaviour. As it immediately starts searching for cycles, it can always return the best found cycle so far, before converging on the optimal. Notably, *S-λD* always found a (possibly non-optimal) solution in few seconds, and oftentimes found the optimal solution a lot sooner than its termination proving that no better solution exists. Even in the volunteer domain, where *S-λD* performs poorly, it finds a decent solution before *CP-MCR* finds any. *CP-MCR* also incrementally improves on the best found ratio, however before being able to do so, it needs to spend a lot of effort to expand the entire

state-space. Figure 3a shows the best ratio found along the search for the most difficult instance of each domain where both algorithms finished. *S-λD* excels in this regard, finding a solution in less than one tenth of a second in most cases, whereas *CP-MCR* sometimes fail to find any solution.

Finally, we analyze the performance of the algorithms when scaling the size of the clock constants. To that end, we fixed a single instance per domain and scaled all constants in the model by a factor. Additionally, we add 1 to the constants to remove large common divisors, which means that the problem can no longer be reduced to the original by simply dividing by the scaling factor. Figure 3 shows that *CP-MCR* suffers a large increase in run-time when scaling the clock constants. In contrast, scaling the clock constants does correlate with the run time of symbolic $\lambda$-deduction. This is not surprising as scaling can result in a polynomial increase in the number of discrete states in the underlying transition system, which has a direct proportional increase in the number of vertices in the doubly weighted graph of *CP-MCR*. Contrarily, scaling the size of the constants does not have a direct impact on the number of symbolic states to explore, and in most cases it only increases the size of the zones. However, adding 1 to the constants after scaling sometimes changes the solution space, thus we also see some fluctuations, especially in the volunteer domain.

## 6 Conclusion

We introduced symbolic $\lambda$-deduction, a novel approach for finding ratio-optimal cycles in cost-reward timed automata. Our approach combines the cost and reward functions into a single weight, circumventing some significant barriers that arise in the doubly priced setting. *S-λD* incrementally improves the best found cycle, searching with priced zones. We proved that it converges on an optimal solution.

Our experiments show that *S-λD* outperforms the baseline, *CP-MCR*, in many cases, except when the symbolic representation is inefficient and leads to a larger number of explored states. However, even in those cases, *S-λD* is far more robust with respect to large values in the temporal constraints.

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
