# OpenReview forum: "Optimal Infinite Temporal Planning: Cyclic Plans for Priced Timed Automata"
_icaps-conference.org/ICAPS/2024/Conference — ICAPS 2024_

### Official Review · Reviewer_fXNg · 2024-01-07

**Significance And Importance:** 2
**Soundness:** 2
**Novelty:** 4
**Clarity:** 3
**Confidence:** 3

**Weaknesses:**

0: Minor weaknesses requiring some work to be addressed for the paper to be accepted.

**Contributions Of The Paper:**

This paper introduces a new method, symbolic λ-deduction (S-λD), for finding ratio-optimal cycles in cost-reward timed automata. This method utilizes the symbolic representation of the pricing zones to compactly explore the state space, which reduces the problem to a single priced automaton and avoids the state space explosion problem in traditional methods. At the same time, new cycles with improved ratio can be found incrementally, guaranteeing an optimal solution at termination, or proving that no better solution exists. Experimental evaluation shows that S-λD outperforms concrete state-space search alternatives in the surveillance and job scheduling domains, with good anytime behavior.

**Ethical Considerations:**

(5) Excellent: The paper comprehensively addresses all of the applicable ethical considerations

**Nomination For Best Paper:**

No

**Overall Evaluation:**

-1: (weak reject)

**Questions For Authors:**

1.	Why does S-λD perform poorly in the volunteer domain compared to the Surveillance and Job Scheduling domains?
2.	Is it possible to add more domains for experimental comparisons to be more convincing?
3.	Is it possible to add more comparisons with existing methods to the experimental evaluation, such as the methods mentioned in the paper Tolonen, French, and Reynolds 2018, David et al. 2011, etc.?

**Reproducibility:**

3: Authors describe the implementation and domains in sufficient detail.

**Strengths Of The Paper:**

1.	The paper introduces a new method (S-λD) for solving the ratio-optimal cycles problem in cost-reward timed automata that yields optimal solutions with very good anytime behavior.
2.	The paper is clearly structured and the language is fluent.
3.	The paper clearly proves the convergence of the proposed method S-λD, which provides a theoretical guarantee for the effectiveness of the method.

**Weaknesses Of The Paper:**

1.	The paper does not discuss in detail the reasons for the poor performance of S-λD in certain domains (e.g., the Volunteer) and possible ways to improve it.
2.	In analyzing the performance of the algorithm when scaling the size of the clock constant, it is not clear exactly why adding 1 to the constant after CP-CRM scaling changes the solution space.
3.	The relevant literature mentioned in the paper, such as Patrizi et al. 2011, David et al. 2011, etc., has relevant methods and domains that can be used in the experiments of this paper. The experimental evaluation of this paper needs to add domains and comparative methods.

---

> ### Author Rebuttal · Authors · 2024-01-28
>
> Thanks for your review. Let us clarify why we disagree with the weaknesses mentioned.
>
> Q1) We analyzed Volunteer in detail. In the paper, with space restrictions, we just explained that s-ld is slower due to exploring more states. The main causes are: (1) There are not many clocks and constants are small, so there are not that many integer points; and (2) there are a lot of solutions with different ratios and the optimal plan tends to be very long so lambda deduction uses many iterations.
> Note s-ld's performance in Volunteer is not that poor. It takes longer to get the optimal solution than explicit (solving 2 tasks less), but it finds suboptimal plans a lot faster. And, it is much better with larger clock constants (Fig 3). It is reasonable to expect that no technique will dominate in all benchmarks and under all metrics.
>
> We'll clarify in the paper that adding +1 is a simple and reproducible way to ensure tasks cannot be simplified dividing all constants by the mcd. The solution space can differ greatly, as happens in Volunteer.
>
> Q2) We consider our evaluation and benchmarks fully adequate to assess the strengths/weaknesses of the approach. More domains could be added but there is not much extra space to discuss the results in detail.
>
> Q3) There are no previous methods or benchmarks suitable for comparison. Neither David etal. or Tolonen etal. made their source code or benchmarks publicly available, their methods are not easily reproducible, and they do not deal with exactly the same problem:
>   - David et al.'s method is only for problems with one clock, so they cannot be applied to our domains. Having 1 clock is a very big limitation to model planning problems with concurrent actions/events. They use a single domain, which is very simple. We reproduced their hardest task for the rebuttal and s-ld solves it in 3ms instead of 77s they reported.
>  - Tolonen et al. is a metaheuristic approach without any optimallity guarantee. They use syntetic randomly generated TA, not representative of planning applications. And they have up to 25 locations, where as our parallel compositions get up to 1000 locations.
>   - Patrizi etal. is a completely different setup, on classical planning, not temporal planning with costs/reward.
>
> We see as a contribution our first benchmark set on temporal planning problems with infinite plans represented as TA with cost/reward: inspired by potential applications, and scalable on different types of objects and the clock constraints.

---

### Official Review · Reviewer_wowC · 2024-01-19

**Significance And Importance:** 3
**Soundness:** 4
**Novelty:** 3
**Clarity:** 4
**Overall Evaluation:** 2
**Confidence:** 3

**Weaknesses:**

2: No major or minor weaknesses.

**Contributions Of The Paper:**

This paper addresses the problem of finding infinite cyclic plans of minimal cost/reward ratio in priced timed automata. The paper develops a technique, called symbolic λ-deduction (S-λD), and evaluate it experimentally on several benchmarks, comparing it against a baseline based on the corner-point abstraction of the problem. The baseline builds the corner-point abstraction (i.e. an abstraction that only consider integer evaluation to clocks) and then searches the plan using existing graph-based algorithms. The paper starts from the corner-point abstraction, but to avoid the state explosion problem, it incrementally searches through the state space by representing plans symbolically.

**Ethical Considerations:**

(1) Not Applicable: The paper does not have any ethical considerations to address

**Nomination For Best Paper:**

Yes

**Questions For Authors:**

N/A

**Reproducibility:**

2: Some details are missing, but the paper still appears to be replicable with some effort.

**Strengths Of The Paper:**

The paper is well written, clear, easy to follow, and addresses an interesting and challenging problem. The contribution is solid, the proposed techniques are clever and effective. The experimental evaluation is convincing. Really a good paper in my opinion.

**Weaknesses Of The Paper:**

The experimental evaluation is convincing, as I said, but on the reproducibility side there are no details on the implementation itself, nor on whether it will be made available for further study.

---

> ### Author Rebuttal · Authors · 2024-01-28
>
> Thansk for your review.
>
> Yes, we will also publish the implementation of our algorithms in Zenodo.

---

### Official Review · Reviewer_wZ46 · 2024-01-22

**Significance And Importance:** 2
**Soundness:** 3
**Novelty:** 3
**Clarity:** 3
**Overall Evaluation:** 2
**Confidence:** 3

**Weaknesses:**

0: Minor weaknesses requiring some work to be addressed for the paper to be accepted.

**Contributions Of The Paper:**

The paper proposes a symbolic method for finding optimal schedules in cost-reward timed automata, an extension of timed automata with costs and rewards. It thus makes prior theoretical results based (non-symbolic) corner-point abstraction more practical. An experimental evaluation demonstrates  the effectiveness of the approach and shows that the symbolic method is superior to the non-symbolic one in terms of performance and scalability.

**Ethical Considerations:**

(1) Not Applicable: The paper does not have any ethical considerations to address

**Nomination For Best Paper:**

No

**Questions For Authors:**

1. Are cost-reward TAs more expressive than priced TAs with a single price?
2. Similarly, can the evaluated domains also be modeled as priced TAs? If so, have you compared your approach to symbolic approaches for priced TAs, in particular (Larsen et al., 2001)?

**Reproducibility:**

4: Authors promise to release code and domains (whichever apply).

**Strengths Of The Paper:**

The paper demonstrates that symbolic methods (in particular priced zones) are applicable to determining negative-weight cycles in cost-reward timed automata, which is a non-trivial contribution. The experimental analysis is well done and demonstrates the general applicability of the approach while also analyzing weaknesses. The paper is generally well-written and the proofs seem to be correct.

**Weaknesses Of The Paper:**

It would be helpful to have a discussion of the differences between priced TAs and cost-reward TAs, e.g., why a cost-reward TA cannot be reduced to a priced TA by setting the cost to the ratio. This would be important in particular because there are symbolic methods for priced TAs and it remains unclear why these are not applicable to cost-reward TAs.

Minor comments:
* In the proof of Theorem 7, there seems to be a missing eta in the definition of v_j.
* Below Corollary 8: 'times-tamps'
* also p 5, right: 'valuationfrom' (missing space)
* p7, right: 'members,but' (missing space)

--------
Post Rebuttal:
The rebuttal has clarified my concerns. It would be important that the authors include a brief discussion comparing priced TAs and cost-reward TAs.

---

> ### Author Rebuttal · Authors · 2024-01-28
>
> Thanks for your review.
>
> Q1) The cost-reward TA are more expressive, because they have ratio optimal solutions that converge for infinite plans. In contrast, in an infinite plan, price TA will either reach a point where no cost is ever incurred again or it will diverge. One can consider cost-time ratio optimal infinite plans for priced automata, however, cost-reward TA are a generalisation of that (we can simulate that by choosing a reward function that simply measures the time passed, but we have other options for reward such as getting reward every time a goal is accomplished).
>
>  This is what we meant to illustrate with our example at the beginning of Section 3 (a cost/reward of 1/1 is not necessarily better or worse than 2/3), since the overall ratio depends on the cost and reward of the entire cycle, this cannot be reduced to a single price. We will clarify this in the paper.
>
> Q2) Note that techniques finding a cost-optimal plan to reach a goal are not directly applicable to find ratio-optimal infinite plans. But indeed, in some sense, we build on the work by Larsen et al., 2001. For each lambda value we construct a priced TA and find negative cycles with similar techniques, except we need to handle negative costs and new criteria for termination.

---

### Meta-Review · Area_Chair_Sf6E · 2024-02-06

**Recommendation:** Accept (Oral)
**Confidence:** 4

**Metareview:**

There is an overall positive consensus about this work, which appears rather novel, interesting, well written and solid. The reviewers raised some issues, which the author response seems to have addressed in a satisfactory way. In case of acceptance, I recommend the authors to suitably incorporate their responses in the paper.

**Ethical Considerations:**

(1) Not Applicable: The paper does not have any ethical considerations to address